# "Help! I need some music!": Analysing music discourse & depression on Reddit

**Bhavyajeet Singh**[1☯]*, **Kunal Vaswani**[1☯], **Sreeharsha Paruchuri**[1], **Suvi Saarikallio**[2], **Ponnurangam Kumaraguru**[1], **Vinoo Alluri**[1]

**1** International Institute of Information Technology, Hyderabad, India, **2** University of Jyväskylä, Jyväskylä, Finland

☯ These authors contributed equally to this work.
* bhavyajeet.singh@research.iiit.ac.in

**Data Availability Statement:** The dataset created and used is made public under the CC BY ("Attribution 4.0 International") license and is available at https://www.kaggle.com/datasets/bhavyajeet/reddit-music-discourse.

## Abstract

Individuals choose varying music listening strategies to fulfill particular mood-regulation goals. However, ineffective musical choices and a lack of cognizance of the effects thereof can be detrimental to their well-being and may lead to adverse outcomes like anxiety or depression. In our study, we use the social media platform Reddit to perform a large-scale analysis to unearth the several music-mediated mood-regulation goals that individuals opt for in the context of depression. A mixed-methods approach involving natural language processing techniques followed by qualitative analysis was performed on all music-related posts to identify the various music-listening strategies and group them into healthy and unhealthy associations. Analysis of the music content (acoustic features and lyrical themes) accompanying healthy and unhealthy associations showed significant differences. Individuals resorting to unhealthy strategies gravitate towards low-valence tracks. Moreover, lyrical themes associated with unhealthy strategies incorporated tracks with low optimism, high blame, and high self-reference. Our findings demonstrate that being mindful of the objectives of using music, the subsequent effects thereof, and aligning both for well-being outcomes is imperative for comprehensive understanding of the effectiveness of music.

## Introduction

Depression can seriously hamper social and intellectual development in youth. As reported by the World Health Organisation (WHO), it is also one of the leading causes of disability in adolescents and young adults globally per year. This calls for effective approaches to address this debilitating condition, especially at an early stage. Music plays an important role in the lives of individuals suffering from depression as one of its key functions is to help regulate mood and emotions regardless of age, gender, ethnicity, or other social factors [1]. Individuals choose varying music and listening strategies to satisfy and reinforce their psychological needs in addition to achieving particular goals, be it relaxation, energizing, diversion, self-reflection, or mood enhancement, amongst others [2].

Emerging empirical evidence has demonstrated that certain musical engagement strategies are associated with measures of ill-health including the risk of depression and antisocial

**Funding:** The author(s) received no specific funding for this work.

**Competing interests:** The authors have declared that no competing interests exist.

behavior, particularly in youth [3–5]. Healthy-Unhealthy Music Scale (HUMS) [6], a self-report questionnaire based on research with depressed adolescents, was developed to identify maladaptive musical engagement that is associated with high psychological distress and risk of depression. Such maladaptive musical engagement involves listening strategies for mood worsening, avoidance, and rumination. To add to this, Miranda and Claes' study on 418 French-Canadian adolescents demonstrated that individuals suffering from depression are inclined to use music as an avoidant coping mechanism which can prove to be maladaptive or lead to non-beneficial outcomes. Individuals suffering from depression do engage in music for a variety of emotion regulation strategies and attempt to reach a more positive mood [7], but music listening ends up evoking memories and experiences of higher negative valence for them than for the healthy controls [8]. It can thus be argued that there is a vicious cycle in which the antecedent moods impact the choice of music and bias the reactions to it, for instance towards using music to ruminate over one's negative states, which is then likely to contribute to further mood-worsening. On a very broad level, emotion regulation through music can be divided into emotional processing of experiences (emotional work, accepting feelings) or moving away from them (searching for a distraction, relaxation, and mood improvement) [9]. A recent music listening experiment by Schäfer et al. [10] showed that music listening helped to relive experimentally induced loneliness and to improve mood both in cases when music was self-selected to be comforting and mood-validating (i.e. promoting emotional processing) and in cases when music was self-selected to be happiness-inducing and distractive away from worries. However, these studies do not reveal what sort of music, in terms of musical and lyrical content, results in potentially negative or positive outcomes. Through self-reports on focus groups, Garrido et al. [11] evidenced that the choice of music, be it mood-congruent or incongruent, may result in both positive and negative outcomes. The authors, however, acknowledge the limitation of self-reports from a selected small sample. Moreover, interview-based settings may lead to bias stemming from demand characteristics and social desirability. Garrido et al. [11] recommend that future studies employ more objective approaches and measures on a larger sample.

Another drawback of prior research linking music listening and depression is the absence of data collection in ecologically valid settings. Surana et al. [12, 13] bridge this gap by collecting 541 individuals' listening histories on Last.fm in addition to self-reports of HUMS and psychological distress scores, which are indicative of susceptibility to depression. Results revealed marked differences in the musical preferences of those at risk for depression when compared to those who weren't. Specifically, a preference for music with low-arousal emotions, especially indicating sadness, typically belonging to dream-pop, neopsychedelic genres, and high repetition in listening patterns were found to be characteristic of individuals with depression risk. While all these studies examined music consumption in more ecologically valid settings, they, however, do not examine the listening goals of individuals. Linking musical content like lyrics, and acoustic features with listening goals could shed more insights on music content that is associated with maladaptive mood regulation.

The social media platform Reddit comprises communities of users who interact via subreddits (https://www.reddit.com/subreddits). User anonymity on Reddit allows individuals to be more open about their experiences on all topics. These subreddits have thus grown to be important resources of social and emotional support for individuals suffering from depression. The r/depression subreddit (https://www.reddit.com/r/depression), for instance, had grown from around 100k members in 2015 to 590k members in 2020 (https://subredditstats.com/r/depression). The vast community on different subreddits offers the prospect of conducting a large-scale study on user behavior analysis [14, 15]. For example, Choudhury et al. [16] built a prediction framework that identifies users who transition from discussing mental health issues

to discussing suicide ideation. They used binary classifiers accompanied by a group of predictor variables like lower readability of posts and separation from the social realm. However, no study has examined music shared on this platform and the social discourse associated with it. Our study is the first to extend prior research on music and depression by examining online social discourse to unearth context and musical content.

To examine online social discourse, it is crucial to use Natural Language Processing (NLP) techniques to deal with the plenitude of available online content. A variety of NLP techniques are available that can aid in sorting and provide meaning to the platform's free (unstructured) content. Especially when it comes to topics like depression, which is marked by linguistic predictors like increased perceptual processes, references to sadness, and discrepancies or greater negative feelings. To elucidate, Tadesse et al. [17], use the posts of users on Reddit to determine any characteristics that may reflect depressive mindsets. To extract various features relating to users' linguistic patterns, they employ Latent Dirichlet Allocation (LDA) [18] to discover a specific set of terms used more commonly by users when it comes to depression. Topics associated with these terms included depression (die, hate, alone), job (boss, boring, fired), friendship (care, happy, best), and broke (find, help, heartbroken). Further, various classification models were used to detect depression-related posts in the forum. Features obtained from LDA were used by Tadesse et al. to enhance the classification performance of a depression detection model. Unlike Tadesse et al. [17] which targeted patterns of specific users, Feldhege et al. [19] used LDA to develop a model of 26 topics to investigate the prominent themes in r/depression. They found that emotional states (guilt, sadness, loneliness) and motivational thoughts (try, work, hard) were two of the most prominent topics in addition to others such as coping and self-reflection. While both studies relied on LDA, this approach does not consider context or word order as it interprets the document's vocabulary as a random collection of words. Another popular strategy for gathering topics in an unsupervised manner from a set of documents is clustering short-text intents using a textual embedding that preserves context. A popular technique that employs this is BERTopic [20], leveraging transformer models [21] which are widely used for their distinguished performance in NLP tasks and a class-based TF-IDF to extract homogenous topics. As mentioned earlier, while music is used to cope with various emotional states in the context of depression, understanding the purpose of music listening and the kind of music to achieve certain emotion-regulation goals is imperative in ensuring positive outcomes. Knowledge on healthy vs. unhealthy patterns can advance early identification of depression risk and serve preventive actions. In our study, first, we analyze the posts and comments within the r/depression subreddit to identify text that captures music listening strategies. Subsequently, we analyze features of music used for these strategies. A distinguishing feature of our study is that it provides, on a large-scale, insight into the several music-based coping strategies individuals use. Furthermore, since the data collected and analyzed has been done without experimenter intervention, it increases the ecological validity of the hence obtained results. In our study, we address the following research questions:

- RQ1: What are the various healthy and unhealthy emotion-regulation goals of music listening in the context of depression?

- RQ2: What kind of musical content (acoustic features and lyrical themes) is preferred to achieve these goals?

To address the RQ1, we gather discourse that links music and depression in subreddits. For RQ2, we analyze the tracks shared in such discussions.

## Materials and methods

The block diagram of the methodology that we used is shown in Fig 1. The steps for data collection, text analysis, and music analysis are outlined below.

### Data collection

A subreddit is a collection of posts from users, about a specific topic, that include a title, description, and comments. While it is not possible to obtain the demographics of a particular subreddit, Statistica (https://www.statista.com), a popular source for obtaining website statistics, reports that the majority of users on Reddit are male and under 50 years of age with most traffic coming from western countries with USA, UK and Canada being the top 3 contributors. However these numbers only represent the users of the entire platform and not specific to the subreddits considered in this study. The r/depression subreddit was used for collecting data since it provides the largest peer support for anyone suffering from a depressive disorder on Reddit with 891k members. The subreddit has been frequently used by individuals to fulfill certain mood-regulation goals. Fig 2 shows an instance of a post demonstrating this. The title of the post, in the given example, reflects users' intent for their musical needs and choices in the context of depression. As a result, the first step in data collection was to accumulate these titles, in the r/depression subreddit, to unearth various associations users have with music. Also, post titles provide shorter sentences that are easier to distinguish compared to other texts in a post like descriptions, comments. Furthermore, we collected songs from the same subreddit each accompanied by a text snippet that describes the context in which the track was recommended. The r/depressionMusic (https://www.reddit.com/r/depressionMusic) subreddit was also investigated to expand the dataset and add more tracks. The data collected from

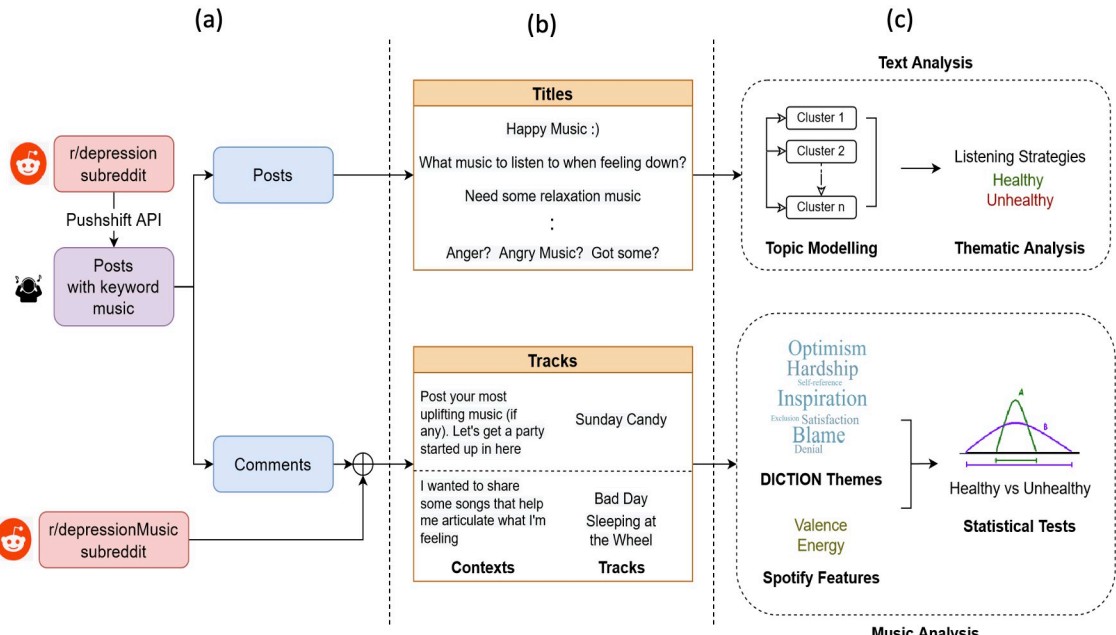

**Fig 1. Block diagram of the methodology, including a procedure of data collection (Text and Music).** Data is being collected from reddit posts using an API (a). Titles reflect a variety of user-submitted posts on music in subreddits. Along with this, instances of context used in combination with the music tracks are collected (b). The data is then processed to generate results using Topic modeling, Thematic analysis and Statistical tests (c).

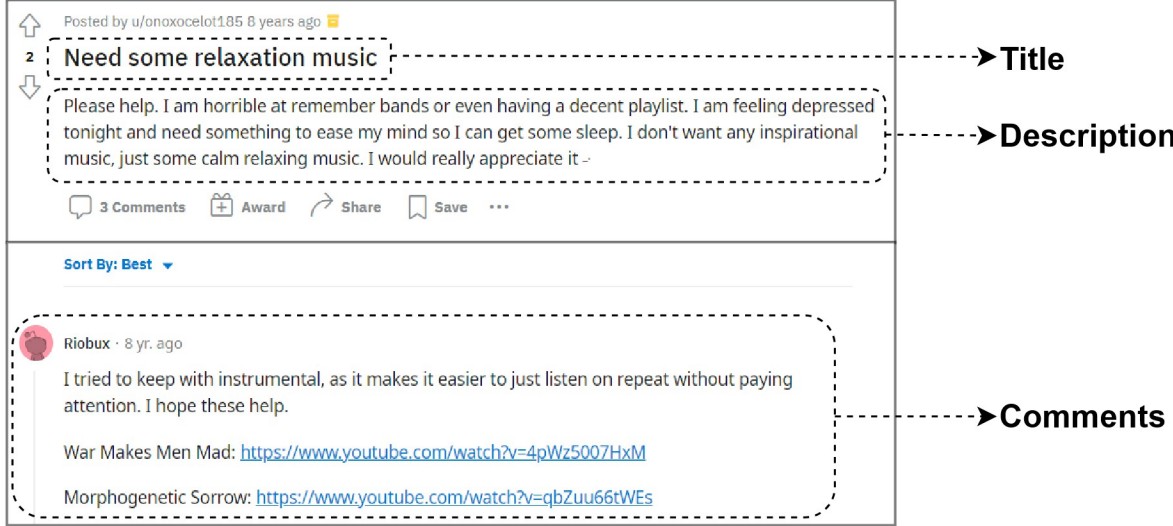

**Fig 2. An example of a Reddit post used in our study with a title, description, and comments.** The title reflects the intention of the user to use music, the description provides further details about the post made by the user and the comments section shows various tracks shared.

Reddit is entirely public and anonymous. We ensure that all the collected data and the process of data collection including the posts, track information and lyrics complied with the terms and conditions for the source of the data.

**Collecting titles.** We have used the Pushshift API [22] to crawl data from the specific sub-reddits, to make use of advanced search functionality to filter entries with a music-related intent. During crawling, the term music was used to filter posts related to music. The API allows different types of queries: searching this keyword in all of the post's data (title, description, comments) or only the title of the post. Our search was restricted to the title of the post and a total of 2,788 posts were collected between 1 Jan 2010 and 1 Jan 2022. Examples of titles from these posts are shown in Fig 2.

**Collecting tracks.** Posts collected above had a general format with a user expressing what they wanted in the title and many users responding in the comments, often with links to music tracks or playlists. Firstly, to gather the post IDs for individual posts, the Reddit API was used. These IDs were utilized to aggregate all of the comments on a particular post using the Python Reddit API wrapper praw(https://github.com/praw-dev/praw). Following this process, a total of 8,688 comments were collected. The first step to examining musical content associated with healthy and unhealthy listening strategies was to accumulate comments that were associated with music sharing. To this end, we handpicked the tracks' information, either submitted in the form of track name and artist, Spotify links, or Youtube links. r/depressionMusic, another support group for exchanging music that helps users go through difficult times or remind them of what they went through, was also used for collecting tracks. A total of 150 tracks were gathered from such comments along with text snippets that led to the recommendation which acted as context. Fig 1(a) depicts the process of collecting the comments using Pushshift API. Healthy-Unhealthy Music Scale (HUMS [6]) was used to inspect the context of the song and thus categorize it into healthy or unhealthy. Table 1 provides examples of these snippets and their categorization.

**Table 1. Examples of text snippets and subsequent categorization into healthy and unhealthy using HUMS.**

| Text snippet utilized as context to categorize Track | Category |
| --- | --- |
| Share music that puts you in a good mood :) My personal favorite artist to put me in a good mood is.. | Healthy |
| I thought we could make a list of music that makes you feel happy or better. | Healthy |
| That being said, im looking for some music to just relax to. so i figured instead of looking | Healthy |
| give me your darkest looking for dark/depressing music to cry myself to sleep to | Unhealthy |
| Like you said it can put you in a bad cycle but I cant help but love it lol. | Unhealthy |
| I have a Spotify playlist for depressing music. Anyone else do this? | Unhealthy |

## Text analysis

Post titles were examined to discover the many reasons why individuals use music in the context of depression. We employed a mixed-methods approach to topic modeling. First, NLP techniques were used to create latent semantic representations of these intents/reasons, which were then clustered to identify different topics. Then, we evaluated these results qualitatively to ensure perceptual validity and organize them via a thematic analysis into broader categories that capture healthy and unhealthy music listening strategies. This is explained in detail in the following sections.

**Topic modeling.** In order to identify underlying themes or topics present in text, we used natural language processing, specifically the BERTopic [20] technique to create dense sentence clusters and generate meaningful topic descriptions for each cluster. In order to do this, the title of the post is first represented as embeddings which are then used to compute similarity with other titles. For this, we used sentence transformers [23] owing to their superior performance in sentence similarity tasks. Sentences which have similar semantic meaning are closer together in this embedding space. We then apply UMAP (Uniform Manifold Approximation and Projection) dimensionality reduction [24] on the embeddings extracted using the Sentence-BERT(Birectional Encoder Representations from Transformers) transformer [25] as a pre-requisite for the clustering algorithm. These vectors are then clustered together into dense clusters using the HDBSCAN clustering algorithm [26]. These vectors are then mapped back to their source sentences, finally giving us clustered Reddit posts. Then, topic descriptions are extracted for each of the generated clusters using the class based tf-idf [27] algorithm. The topic description includes the most prevalent n-grams found in the sentences. The representative n-grams for the clusters are extracted using a statistical method, however, the embeddings used for clustering are contextual in nature. This allows the clustering algorithm to utilise deeper semantic meanings while grouping the posts. This pipeline is depicted in Fig 3.

**Thematic analysis.** Thematic analysis entailed qualitative evaluation of the topic descriptions corresponding to each of the generated clusters followed by organization of these topics into coherent themes and finally into broader categories on the basis of prior literature. This procedure is explained in what is to follow. We start by examining the set of n-grams obtained as a result of topic modeling. These n-grams act as a preliminary label for each cluster. Subsequently we validate the representatives for each cluster by qualitatively analysing each of the sentences pertaining to a cluster. In order to further organise these clusters into themes, all the sentences comprising each topic cluster were subjected to qualitative content analysis. Finally these themes were organised into broader categories representing healthy or unhealthy listening strategies. The organization was primarily based on the HUMS scale [6]. Along with HUMS, various other research noted in aforementioned literature were also explored. This mixed-method approach to organizing Reddit posts benefits from the feasibility of organizing

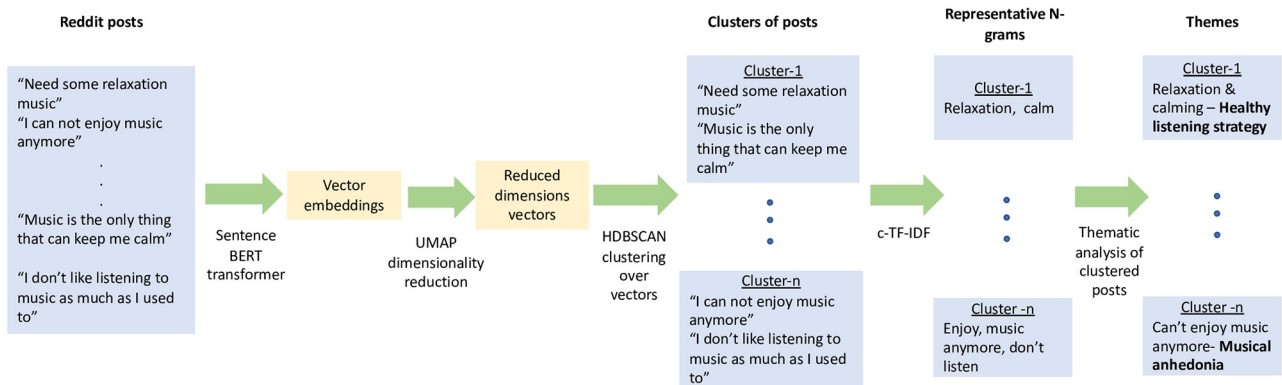

**Fig 3. Topic modelling pipeline used for clustering the Reddit posts.** All Reddit posts are clustered into different themes using the BERTopic technique. This picture illustrates with examples, the different components of the pipeline.

huge corpora afforded by the quantitative method while retaining perceptual relevance and validity afforded by the qualitative evaluation.

## Music analysis

The lyrics corresponding to each of the tracks were extracted using the lyrics Genius. API (https://docs.genius.com) Since lyrics weren't obtainable for all of the tracks, 84 tracks out 150 were eventually used (39 for Healthy and 45 for Unhealthy). Next, we used DICTION software (https://dictionsoftware.com) to extract lyrical themes. DICTION relies on dictionaries and a list of 35 predefined topics to determine the content of any given text of social discourse. Among the 35 variables, we chose features that are associated with cognitive models of depression [28–30]. These models contend that the process of depressive cognition is associated with high Blame, Pessimism, and negative Self-reference. Hence, the DICTION themes evaluating Self-reference, Blame, Optimism, and Hardship were chosen. Also, since lower life satisfaction, lack of motivation, and escapism are known to be common in such individuals [6], we additionally pick Satisfaction, Inspiration, Exclusion, and Denial. While the DICTION categories are self-explanatory, we provide detailed descriptions of these chosen themes in the Supplementary material. For each track, we obtained frequency scores for the occurrence of words in the dictionary lists for the eight chosen lyrical themes. For acoustic features, the tracks were mapped to their Spotify links for feature extraction. We use the Spotify developers API (https://developer.spotify.com) to extract track related features for all the tracks in our dataset. These features included valence, which represents positiveness of a track, and energy, which is a perceptual measure of intensity and activity of a track. To analyzte differences in musical content between healthy and unhealthy strategies, we performed statistical tests of difference, namely, a two-tailed Mann-Whitney U (MWU) [31] Test on the features of the tracks associated with the respective strategies.

## Results

### Text

Topic modeling resulted in the formation of 45 clusters. From these clusters, 26 unique themes were identified and grouped into four broader categories using thematic analysis. Table 2 shows the results obtained from thematic analysis of the identified topics along with the

**Table 2. Results obtained from topic modeling and thematic analysis.**

| Theme | Topic n-grams | Frequency |
|---|---|---|
| **Healthy Music Listening Strategies** | | |
| Relaxation / Calming | relaxing, anxiety, sleep, stress, relax, relief | 71 |
| Music to heal | healing, healing music, solace, medicine, heals, music heals | 34 |
| Coping mechanism | cope, coping, music to cope, mechanism, coping mechanism | 50 |
| Music as therapy | therapy, music therapy, therapy music, my therapy, is my therapy | 34 |
| Mood regulation | feel, you feel, feel better, feelings, makes, better | 62 |
| Music makes me happy | happy, happy music, makes me, me happy, music makes me | 40 |
| Music videos | video, this music video, this music, official music video | 39 |
| Sharing music | share, share music, we, to share music, sharing | 26 |
| Music saved life | saved, saved my life, saved my, my life, life, life music | 22 |
| Music keeping me alive | alive, me alive, keeping me alive, keeping me, keeping | 27 |
| Catharsis / Makes me cry | cry, me cry, crying, makes me cry, to music, crying to music | 19 |
| **Unhealthy Music Listening Strategies** | | |
| Mood worsening | worse, music makes, makes, feel, it worse, hurts | 42 |
| Depressive music | depressed, does, anyone, else, listen, depressive | 55 |
| Listening in the dark | to music, listen to music, bed, listening to music, dark | 42 |
| Listening to sad music | sad, to sad music, to sad, sad music, listening to sad | 25 |
| Music as an escape | escape, escape music, an escape, music as, as an escape | 20 |
| Escape to better times | nostalgic, nostalgic music, old music, reminds, reminds me | 21 |
| Addicted to music | else, anyone else, addicted to music, hate, addicted to | 28 |
| Only enjoyment is music | only, the only, thing, the only thing, music is the, care about | 22 |
| **Musical Anhedonia** | | |
| Can't enjoy music anymore | enjoy, enjoy music, music anymore, music anymore don't, don't | 40 |
| Can't listen to music anymore | don't listen, to music, don't listen to, anymore, to music anymore | 18 |
| Lost interest in music | interest, interest in, interest in music, lost, lost interest | 19 |
| **Music Seeking** | | |
| Music recommendations | music recommendations, recommendations, recommendations music | 26 |
| Suggestions | suggestions, music suggestions, suggestions music suggestions | 24 |
| Generic—Music | music music music, music, thread music, music thread | 20 |
| Generic—New Music | new music, need music, need, new, music need music | 18 |

Each category is allotted a Theme, Topic n-grams, and Frequency (number of sentences assigned to the category by BERTopic). comprising the cluster; unigrams, bigrams, and trigrams were used for this. BERTopic also has the advantage of not forcing sentences into clusters, thereby filtering noise.

representative n-grams and frequencies of each of the clusters. As described in the methodology section, the clusters are created based on contextual information whereas the representative n-grams are generated using a statistical method. This explains that while words like "stress" or "anxiety" from the "topic n-grams" column of Table 2 might seem contradictory to the theme of "relaxing/calming", examining the context in which these words were used provides greater clarity. The aforementioned cluster contained posts like "Music that ease anxiety?" or "Relaxation music for deep sleep and stress relief" which fit well with the assigned theme.

For each unique theme, the most representative cluster, that is, the cluster that has the highest proportion of posts that reflects the label is listed in Table 2. The other 19 clusters either resulted in highly overlapping themes with the ones presented here or contained generic posts with little or no context regarding the music listening or sharing strategy and hence have been

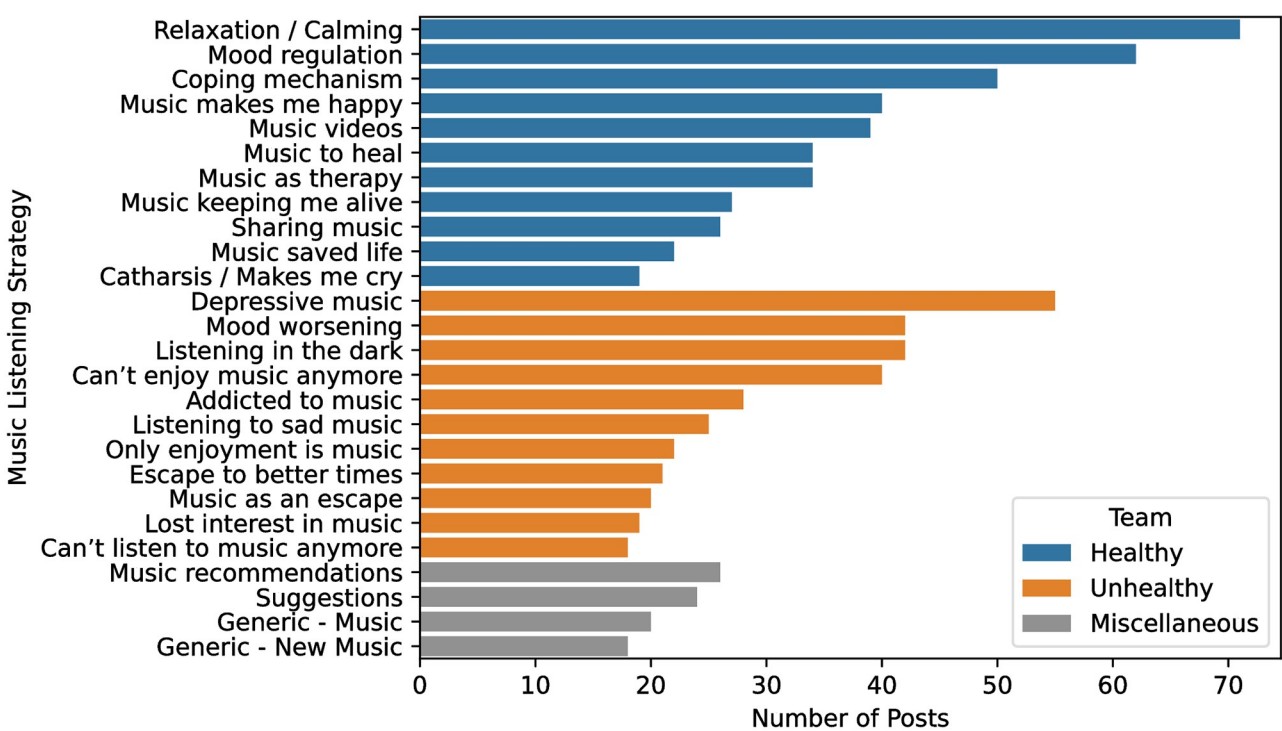

**Fig 4. Distribution of the number of posts assigned to each listening strategy identified in the r/depression subreddit.** Use of music for relaxation and calming is the most common listening strategy overall whereas deliberately listening to depressive music is most common among the unhealthy music listening strategies.

excluded from the table. The entire dataset has been made public and can be found at: https://www.kaggle.com/datasets/bhavyajeet/reddit-music-discourse.

In the Healthy category, a wide range of themes were discovered like Relaxation (relaxing, anxiety, sleep), Music to heal (healing, healing music, solace), Self-reflection (feel, you feel, feel better), Music makes me happy (happy, happy music). Each of the assigned themes had an average of 38 posts, Relaxation incorporating the highest with 71 posts. On the other hand, the Unhealthy category included distinctive themes such as Mood worsening (worse, music makes), Depressive music (depressed, does, anyone), Listening in the dark (to music, listen to music), Music as an escape (escape, escape music). The themes had an average of 32 posts in the Unhealthy category, Depressive music comprising the most with 55 posts. We also identified two additional categories: Musical Anhedonia and Music Seeking. Musical Anhedonia involved titles reflecting individuals who were failing to enjoy music and Music Seeking was associated with titles related to generic music recommendation/suggestion posts.

Fig 4 highlights the distribution of posts assigned to the range of themes in the Healthy, Unhealthy, and Miscellaneous categories. Themes representing musical anhedonia were placed under unhealthy strategies. Overall, a total of 424 posts were assigned to the Healthy category, 332 posts were assigned to Unhealthy, and the 88 posts of musical seeking were assigned to Miscancellaneous. As mentioned before, not all sentences were assigned a cluster, resulting in a group of outlier sentences. A qualitative analysis of these sentences revealed that these were general sentences which did not convey sufficient information regarding the listening context, and hence were not included in the table.

## Music

Violin plots in Fig 5 illustrate the results of the Mann-Whitney U (MWU) Tests conducted on frequency scores for the lyrical themes of the tracks. As can be seen, among the eight chosen themes, the MWU tests revealed a significant difference in Self-reference, Optimism, and Blame (p = 0.0007, 0.0300, and 0.0118 respectively). Higher blame and self-reference, and lower optimism were found in the lyrics from tracks associated with unhealthy music listening strategies. Fig 6 shows the results of MWU tests on the acoustic features of the tracks. Valence of the tracks associated with unhealthy listening strategies was found to be significantly lower than that of the tracks associated with healthy listening strategies (p-value = 0.0067). No significant difference was observed for Energy values of the two groups.

## Discussion

Our study utilizes posts in an online depression forum on Reddit to discover the various intentions behind music listening in the context of depression. The music discourse uncovered on this social media platform displays various strategies that have already been talked about in previous research conducted through surveys and experiments [3, 32]. Our findings provide further detail to these previously identified strategies in an ecologically valid context, allowing for new insights particularly to the perceived healthy and unhealthy aspects of musical emotion regulation in the context of depression, and introduce links between these posts and the musical (acoustic and lyrical) content of the tracks that the users listen to.

## Reddit posts

The Reddit posts, as one would expect, provide a view of music listening as a versatile tool for healthy emotion regulation in the context of depression. Numerous posts describe music as a

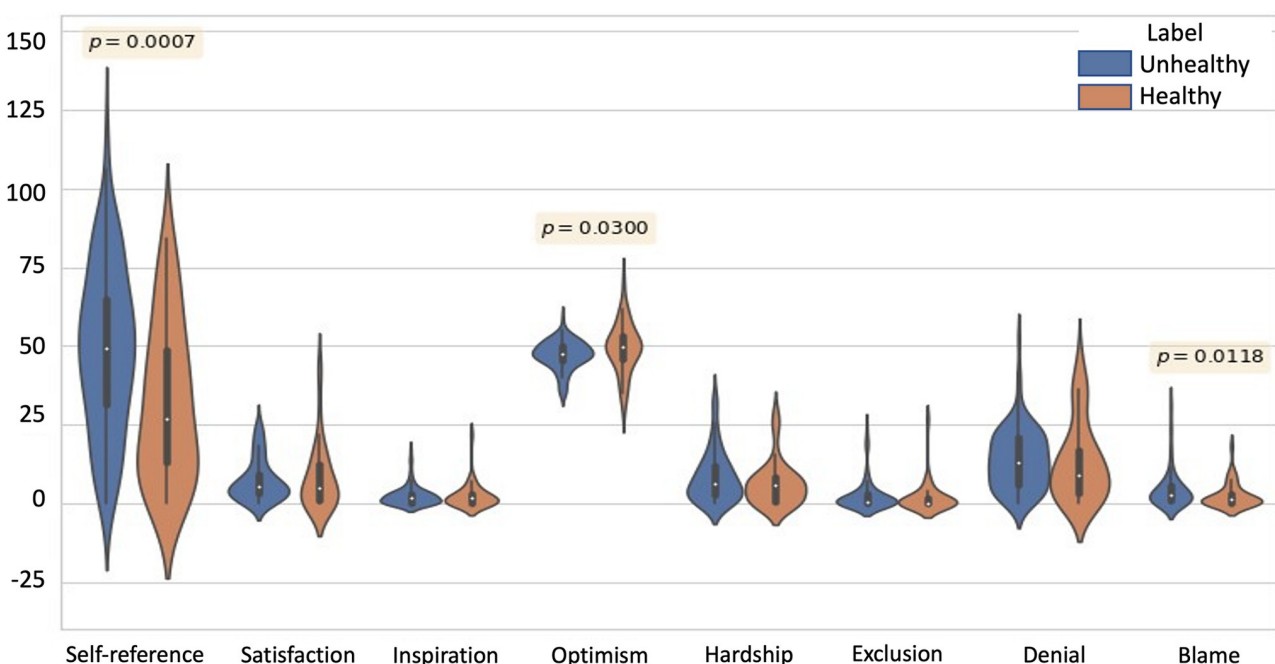

**Fig 5. Violin plots for frequency scores of all the lyrical themes.** The significant differences are depicted using the p-values obtained from the MWU test, in Self-reference, Optimism and Blame. The y-axis represents frequency scores for the occurrence of words in the dictionary lists for each of its semantic variables.

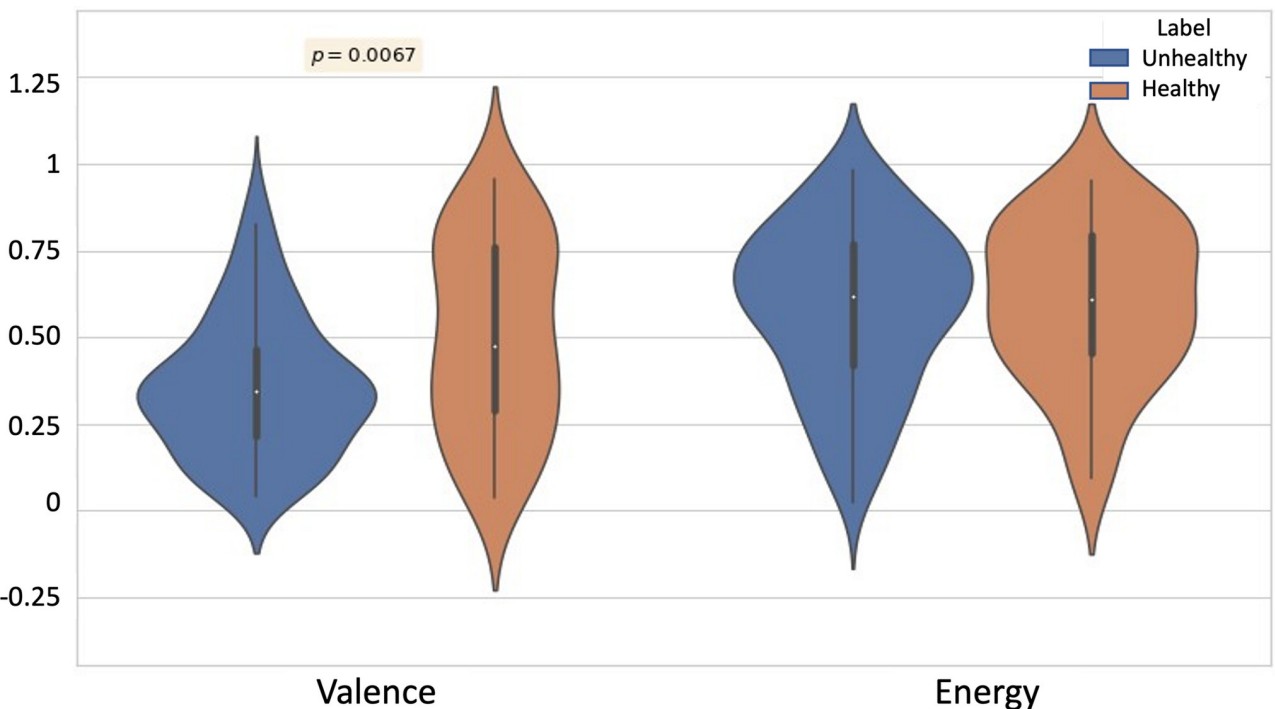

**Fig 6. Violin plots describing the distribution of valence and energy values for the tracks.** Tracks associated with unhealthy music listening strategies have a significantly lower valence when compared to those associated with the healthy listening strategies.

source of healing, therapy, coping, and mood regulation. Music was particularly often perceived to improve mood and to help to relax and these experiences are directly comparable to the Healthy HUMS items like Music helps me to relax, I feel happier after playing or listening to music. The general idea of music as a tool for coping and mood regulation has also been widely discussed in the broader music psychology literature [33–35]. While this work is conducted outside the specific context of depression, many similarities exist in terms of how music can serve as an emotional resource during difficult times. For instance, the Brief Music in Mood Regulation (B-MMR) [36] scale assesses strategies such as diversion (When I feel bad, I try to get myself in a better mood by engaging in some nice, music-related activity), mental work (Music has helped me to work through hard experiences), and solace (When everything feels bad, music understands and comforts me).

In the current data, the topic that received the highest number of nominations was relaxation. Relaxation has been observed to be the most common emotion-related reason for music listening also in general in prior research [37]. As a contrast to the general population-based research, a distinctive topic in our data were the experiences of music as a life-safer, something that keeps one alive, described in topics Music saved life, Music keeping me alive. These are strong expressions that connect music with deep personal value. This is somewhat in line with prior research noting that music becomes emotionally meaningful particularly in difficult times of life [38].

Another distinctive feature were the cathartic experiences of crying, indicated in topics like Catharsis / Makes me cry. These experiences are somewhat reflective of Garrido and Schubert's work [32] on the 'Like Sad Music Scale' which addresses the numerous factors underlying a preference for sad music, including adaptive behviours and participants enjoying sad emotions created by sad music. Finally, we also included experiences of music sharing and

music videos into the healthy music engagement. They are somewhat reflective of the HUMS healthy item Music helps me to connect with other people who are like me, in a sense that they reflect the social potential of the online community. However, they also seem to refer to a somewhat lighter, mundane use of music in terms of providing interesting content and ideas. These experiences then further link to the Miscancellaneous topics of music seeking (Music recommendations, Suggestions, Generic—Music, Generic—New Music) that were not categorized under healthy or unhealthy uses, but which all reflect this allowance of the community for sharing experiences, ideas, and related music.

Perhaps the clearest examples of unhealthy uses of music in our data relate to music listening strategies that are associated with rumination. This is reflected in topics including Mood worsening, Addicted to music, Listening in the dark, Depressive music, and Listening to sad music. These posts paint a picture of music listening that occurs in a dark place with music being perceived as sad and depressive and the listeners ends up feeling worse, yet addicted to doing this. The use of music for rumination is in line with the contents of the HUMS scale [6] with items such as When I listen to music I get stuck in bad memories, When I try to use music to feel better I actually end up feeling worse.

Another unhealthy emotion regulation or coping strategy, avoidance, is reflected in the topics of Music as an escape, Escape to better times. These experiences are illustrative of how music and overindulgence in nostalgia can act as excuses to avoid facing the current situation and related feelings. Such experiences are in line with the work of Miranda and Claes [3] who demonstrated that music listening for avoidance/disengagement, to evade thinking about issues that necessitate immediate reflection, is connected with higher depression levels. These types of experiences are also present in the HUMS scale, with items such as I hide in my music because nobody understands me, and it blocks people out.

Overall, the posts placed in the Unhealthy category are greatly in line with prior research on unhealthy music listening. In the descriptions of their experiences, the listeners seem to observe that the music they listen to is depressive and making them feel worse, but they still turn to it, perhaps due to not having any better remedies either. In addition, the current study also reached some novel observations such as a reduced interest in music and a feeling that music is no longer helpful.

Topics assigned under the category of musical anhedonia were Can't enjoy music anymore, Can't listen to music anymore, Lost interest in music. Anhedonia is a key symptom of severe depression and it can be described as an inability to feel pleasure in activities that were previously considered pleasurable [39, 40]. Posts such as "even music isn't helping now", "Every time I try cope with music, it just feels like it's nothing but noise in my ears.." depict stories of people who used to music as an aid, but have now reached a stage where it has stopped helping. Our findings about the inability to enjoy music reflect such experiences and they may be especially typical in severe cases. These findings resonate with the findings of Carlson et al (2021) [41] who found that participants who begun to experience music negatively during the pandemic also scored high in anxiety. Furthermore, Schrader (1997) [42] have shown significant correlations between general anhedonia and a risk of depression. Hein et al (2022) [43], further explored "the relationship between recent suicidal ideation and the different anhedonias", including musical. Indeed, based on previous works, it could be hypothesized that the inability to enjoy music anymore might be a marker of depression.

## Musical and lyrical features

Content-wise, we further analyzed how the musical and lyrical features of the tracks that the users listen to correspond to the Healthy and Unhealthy strategies discussed above. As could

be expected, the valence score obtained from the acoustic features was lower for the unhealthy posts.

Results of the lyrical content of the music revealed that higher optimism was found in lyrics of tracks associated with healthy strategies. These mirror the results observed in the study conducted by Garrido et al. [5]: lyrics that contain hopeful and motivating, hence optimistic lyrics were found to be associated with positive mood-regulation outcomes. On the other hand, higher blame and self-reference in lyrics of tracks associated with unhealthy strategies suggests that such themes may indeed not be recommended as they may lead to negative outcomes. This is somewhat in line with Sakka's (2020) [8] findings that depression is related to negative self-referential memories being evoked by music listening. Ruminative brooding, which is conceptualized as persistent self-critical nature are key maladaptive processes that need to be targeted for effective treatment [44]. Our work hence suggests that music with lyrics portraying high blame and self-reference is not recommendable for depressive individuals.

To conclude, while much of previous music research has focused on music being a a great resource for mental well-being, pleasure, and self-regulation, some works have also explored the possibility of certain music habits being maladaptive in nature [45]. The findings from our work yet again show that unhealthy music listening and sharing is just as prevalent and highly concerning. People in the past have focused on how music can help with individuals suffering from depression, but it also becomes necessary to discuss what should not be done. By looking at the unhealthy and anhedonic music listening patterns in our study, we can also infer that individuals need to be concerned about not just what they are listening to but also the how and the why.

## Limitations and future work

Although we limited our analyses to r/depression for topic modeling, this work can be further extended to include other smaller depression-related subreddits. In addition, examining other mental health-related subreddits would provide insights into the role of music in other psychiatric and psychological conditions. This study employed a mixed-methods approach, of technology-aided manual annotation to examine social discourse on music and depression. Though the state-of-the-art language models were used to analyze, they come certain inherit limitations. For example, previous works in the area of NLP [46] have shown that transformer-based models face difficulties in understanding negation in language. This was also observed in our work, when analyzing the clusters which denoted anhedonia. In such cases, manual intervention proved to be beneficial and necessary. Another limitation is the use of acoustic features via the Spotify API which is not exhaustive by any means. Extending this work to include other perceptually relevant acoustic features in addition to descriptive tags obtained from Last.fm would provide more insight into the nature of music shared in the context of depression. Nevertheless, the findings obtained are in-line with the previous research done with big music data and smaller lab-based studies.

While user anonymity on Reddit enables users to be more forthright about their experiences on any topic especially those associated with taboo, it does not share users' personal information. Consequently, the research stops short of studying the relationship between music listening behavior and individual traits, which has been done by prior studies. Individual differences such as Personality, Empathic traits [47], or Gender [2] may indeed be significant modulators of music listening strategies.

Though this study highlights interesting patterns in the usage and sharing of music in the context of depression, we must note that these insights are correlational and not causal in nature. Despite its limitations, the study adds to our understanding of various adaptive and

maladaptive music listening behaviors. Such information could then be employed for the timely identification of depressive symptoms, therapeutic systems, and customized recommendation engines.

## Author Contributions

**Conceptualization:** Bhavyajeet Singh, Kunal Vaswani, Suvi Saarikallio, Ponnurangam Kumaraguru, Vinoo Alluri.

**Data curation:** Bhavyajeet Singh, Kunal Vaswani, Sreeharsha Paruchuri.

**Formal analysis:** Bhavyajeet Singh, Kunal Vaswani.

**Methodology:** Bhavyajeet Singh, Kunal Vaswani, Sreeharsha Paruchuri, Suvi Saarikallio, Vinoo Alluri.

**Supervision:** Ponnurangam Kumaraguru, Vinoo Alluri.

**Visualization:** Sreeharsha Paruchuri.

**Writing – original draft:** Bhavyajeet Singh, Kunal Vaswani.

**Writing – review & editing:** Suvi Saarikallio, Ponnurangam Kumaraguru, Vinoo Alluri.

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
