## [Decision Letter · Decision Letter 0]

27 Mar 2023

PONE-D-23-04461“Help! I need some music!”: Analysing music discourse & depression on RedditPLOS ONE

Dear Dr. Singh,

Thank you for submitting your manuscript to PLOS ONE. After careful consideration, we feel that it has merit but does not fully meet PLOS ONE’s publication criteria as it currently stands. Therefore, we invite you to submit a revised version of the manuscript that addresses the points raised during the review process.

We look forward to receiving your revised manuscript.

Kind regards,

Michal Ptaszynski, PhD

Academic Editor

PLOS ONE

Journal Requirements:

2. In your Methods section, please include additional information about your dataset and ensure that you have included a statement specifying whether the collection and analysis method complied with the terms and conditions for the source of the data.

Reviewers' comments:

Reviewer's Responses to Questions

**Comments to the Author**

1. Is the manuscript technically sound, and do the data support the conclusions?

Reviewer #1: Yes

Reviewer #2: Yes

Reviewer #3: Yes

2. Has the statistical analysis been performed appropriately and rigorously? 

Reviewer #1: I Don't Know

Reviewer #2: Yes

Reviewer #3: I Don't Know

3. Have the authors made all data underlying the findings in their manuscript fully available?

Reviewer #1: Yes

Reviewer #2: Yes

Reviewer #3: Yes

4. Is the manuscript presented in an intelligible fashion and written in standard English?

Reviewer #1: Yes

Reviewer #2: Yes

Reviewer #3: Yes

5. Review Comments to the Author

Reviewer #1: This paper - “Help! I need some music!”: Analysing music discourse & depression on Reddit - reports on work that is at the forefront of online data collection methods in this area. With that in mind, most of the comments below are provided with the intention of helping to increase the impact of the work.

With this in mind, it could be helpful to be more explicit about what each step included. For example when describing Topic Modelling, is it possible to give more information about how this worked. Perhaps an example of going through the process would help. It might be good to illustrate what the clusters were of (words, segments, sentences) and how these were analysed thematically. That might help explain how “anxiety” is in “calm” and why “mechanism” is presented as a separate term in “coping mechanism”. Similarly, it might be helpful to provide an example of the thematic analysis in action. Finally, it might be helpful to provide quotes associated with each of the themes so that readers have an idea of the richness of the data underlying the findings.

If you’d like people who are interested in the psychology of music but may not know about big data techniques, then I’d illustrate the technical process more fully. If your main audience is other researchers working in big data, then much of this would presumably be more familiar. Connected to this, there are several technical terms which may not be familiar to readers from fields like music psychology or music and health (such as "latent semantic representations"). It may be worth how many of these you explain or at least point readers to sources where they could find out more. With that in mind, it might be worth checking throughout for readability across the different audiences you would like to engage.

The use of Reddit is an interesting idea with a lot of potential for collecting a lot of data. As the authors point out, it is not possible to know about the identity of participants. Is there any way in which readers can get a sense of who typically might use reddit (in terms of geography, age, etc.) and what that might say about what can (or can’t) be learnt from the study.

The part of the title before the colon looks like it is a quote. Is this from the dataset? If so, it might be good to use it as part of one of the examples or have it feature somewhere in the text. If it isn’t from the dataset, consider modifying the title.

I look forward to seeing this work in print.

Reviewer #2: This research seeks to examine the relationship between music listening strategies and their impact on the well-being of individuals with depression. The authors argue that while music can play an important role in regulating mood and emotions, using music as a coping mechanism can lead to adverse outcomes such as anxiety and depression. They use a mixed-methods approach, including natural language processing and qualitative analysis, to identify healthy and unhealthy music-listening strategies used by individuals on Reddit. The authors also review previous studies on music engagement strategies associated with depression, the limitations of prior research, and the potential of Reddit as a data source.

The introduction is well-structured, informative, and lays a solid foundation for the study. The authors provide extensive background information and highlight the research gap, which adds significance to their study. Overall, the introduction to the paper is well-written and effectively conveys the importance of the research question. Moreover, the method they use to investigate this issue is clever and well executed — looking as it does at the music people choose to listen to on Spotify and examining the relationship between the lyrical content of this and their mood as gauged through their posts on Reddit.

Although it has some obvious limitations, I think the patterns that the study identifies are very interesting — and certainly worth reporting and reflecting on. As the authors note, some of these are to do with the gross nature of the measures. However, the one obvious limitation (that the authors don’t really engage with) is that the study is correlational and hence can’t tease out cause and effect (does negative affect dictate musical preferences or the other way round?). However, as the authors note the study does provide insight into the way that mood and behaviour go together and hence is a useful — and to my mind really rather interesting — contribution to the literature.

6. PLOS authors have the option to publish the peer review history of their article (what does this mean?). If published, this will include your full peer review and any attached files.

Reviewer #1: No

Reviewer #2: **Yes: **Alex Haslam

Reviewer #3: No

---

## [Author Response · Author response to Decision Letter 0]

11 May 2023

The comments by the reviewers have been extremely helpful in guiding our revision process. We agree with all the points raised by the reviewers and have restructured our manuscript accordingly. Below we provide a detailed discussion of the reviewers’ comments and our responses. The text included in >>> TAGS <<< represents the comment from the reviewers which is followed by our responses. 

Reviewer #1

>>> This paper - “Help! I need some music!”: Analysing music discourse & depression on Reddit - reports on work that is at the forefront of online data collection methods in this area. With that in mind, most of the comments below are provided with the intention of helping to increase the impact of the work.

With this in mind, it could be helpful to be more explicit about what each step included. For example when describing Topic Modelling, is it possible to give more information about how this worked. Perhaps an example of going through the process would help. 

In order to explain the procedures more explicitly, we have rewritten parts of the methodology section. Specifically, we have defined topic modelling in further detail with an additional diagram explaining the entire pipeline. 

>>>It might be good to illustrate what the clusters were of (words, segments, sentences) and how these were analysed thematically. That might help explain how “anxiety” is in “calm” and why “mechanism” is presented as a separate term in “coping mechanism”. Similarly, it might be helpful to provide an example of the thematic analysis in action. Finally, it might be helpful to provide quotes associated with each of the themes so that readers have an idea of the richness of the data underlying the findings. <<<

We have explained the topic modelling section with greater clarity now (please refer to the comment above) and we believe that this would help the readers get a better idea of the entire process. Topic descriptions provide the most prevalent n-grams from the clustered sentences.

‘Anxiety’ is an n-gram in the “relaxation/calm” cluster due to the presence of example posts like the following in the corresponding clusters. 

“Music that ease anxiety?”

“Relaxing/therapeutic music for depression and anxiety”.

“Overcome anxiety. Relaxing music, tested method”

We have added the following in the manuscript to clarify the procedure:

“The representative n-grams for the clusters are extracted using a statistical method, the embeddings used for clustering are contextual in nature. This allows the clustering algorithm to utilise deeper semantic meanings while grouping the posts.” …. 

“As described in the methodology section, the clusters are created based on contextual information whereas the representative n-grams are generated using a statistical method. This explains that while words like ‘stress’ or ‘anxiety’ from the ‘topic n-grams’ column of table 2 might seem contradictory to the theme of ‘relaxing/calming’, examining the context in which these words were used provides greater clarity. The aforementioned cluster contained posts like ‘Music that ease anxiety?’ or ‘Relaxation music for deep sleep and stress relief’ which fit well with the assigned theme.”

Similarly, since ‘mechanism’ also appeared as a separate n-gram (without being preceded by ‘coping’), it is present in the list of representative n-grams given by the model. This can be seen in the following examples : 

“Music as a mechanism to cope with life”

“Anyone else use music as a mechanism to cope?”

Furthermore, the entire dataset has been now made public to further provide the exact quotes associated with each of the themes. This would provide the readers a better picture of the actual quotes that are associated with the identified themes in the study. 

>>>If you’d like people who are interested in the psychology of music but may not know about big data techniques, then I’d illustrate the technical process more fully. If your main audience is other researchers working in big data, then much of this would presumably be more familiar. Connected to this, there are several technical terms which may not be familiar to readers from fields like music psychology or music and health (such as "latent semantic representations"). It may be worth how many of these you explain or at least point readers to sources where they could find out more. With that in mind, it might be worth checking throughout for readability across the different audiences you would like to engage.<<<

We would certainly hope our work reaches a wider audience. Hence, we have restructured the methodology section to improve the clarity of our procedure. We have rewritten the procedures using simpler terminology and made sure that relevant references are provided for all technical terms and assumptions. The topic modelling procedure is now illustrated using a more detailed and understandable diagram.

>>> The use of Reddit is an interesting idea with a lot of potential for collecting a lot of data. As the authors point out, it is not possible to know about the identity of participants. Is there any way in which readers can get a sense of who typically might use reddit (in terms of geography, age, etc.) and what that might say about what can (or can’t) be learnt from the study. <<<

We agree that there is a lack of information regarding the identity or even the demographics of the participants. However as mentioned in the paper, reddit is an anonymous platform and does not provide personal information about the users. 

Statistica is a popular source for obtaining statistics such as traffic on a particular website and other user demographics including age and gender distribution but the source of those numbers is unclear for Reddit. According to Statistica, the majority of users on Reddit are male and under 50 years of age with most traffic coming from western countries with USA, UK and Canada being the top 3 contributors. 

However these numbers only represent the users of the entire platform and not specific to the subreddits considered in the study. 

We have now added the following in the Data Collection Part of the manuscript:

“While it is not possible to obtain the demographics of a particular subreddit, Statistica, a popular source for obtaining website statistics, reports that the majority of users on Reddit are male and under 50 years of age with most traffic coming from western countries with USA, UK and Canada being the top 3 contributors. However these numbers only represent the users of the entire platform and not specific to the subreddits considered in the current study.”

We have also addressed this in the Limitations and Future Work section by rephrasing the following sentences:

“While user anonymity on Reddit enables users to be more forthright about their experiences on any topic, especially those associated with taboo, it does not share users’ personal information. Consequently, the research stops short of studying the relationship between music listening behaviour and individual traits, which has been done by prior studies. Individual differences such as Personality, Empathic traits, or Gender among other demographic variables may indeed be significant modulators of music listening strategies.”

>>>The part of the title before the colon looks like it is a quote. Is this from the dataset? If so, it might be good to use it as part of one of the examples or have it featured somewhere in the text. If it isn’t from the dataset, consider modifying the title.<<<

That part of the title has been used as a reference to the 1965 Beatles track titled “Help!”. The exact phrase does not appear in our dataset, however close variations from various contexts like “Music helps so much”, “I need music”, or “I need help with my music addiction … Please help!” do exist in our dataset.

>>>I look forward to seeing this work in print.<<<

We are thankful for your comments. They have helped us in improving the quality of the manuscript. 

Reviewer #2

>>>This research seeks to examine the relationship between music listening strategies and their impact on the well-being of individuals with depression. The authors argue that while music can play an important role in regulating mood and emotions, using music as a coping mechanism can lead to adverse outcomes such as anxiety and depression. They use a mixed-methods approach, including natural language processing and qualitative analysis, to identify healthy and unhealthy music-listening strategies used by individuals on Reddit. The authors also review previous studies on music engagement strategies associated with depression, the limitations of prior research, and the potential of Reddit as a data source.

The introduction is well-structured, informative, and lays a solid foundation for the study. The authors provide extensive background information and highlight the research gap, which adds significance to their study. Overall, the introduction to the paper is well-written and effectively conveys the importance of the research question. Moreover, the method they use to investigate this issue is clever and well executed — looking as it does at the music people choose to listen to on Spotify and examining the relationship between the lyrical content of this and their mood as gauged through their posts on Reddit.

Although it has some obvious limitations, I think the patterns that the study identifies are very interesting — and certainly worth reporting and reflecting on. As the authors note, some of these are to do with the gross nature of the measures. However, the one obvious limitation (that the authors don’t really engage with) is that the study is correlational and hence can’t tease out cause and effect (does negative affect dictate musical preferences or the other way round?). However, as the authors note the study does provide insight into the way that mood and behaviour go together and hence is a useful — and to my mind really rather interesting — contribution to the literature.<<<

We thank Reviewer 2 for the positive comments.

We agree that our study is correlational and hence can’t tease out cause and effect. In the manuscript we do not claim that this study can be used to interpret cause or effect, however we have now explicitly included this as one of the limitations in the ‘limitations’ sections of the paper. 

This information is presented in the paper as follows: 

“While this study highlights interesting patterns in the usage and sharing of music in the context of depression, we must note that these insights are correlational and not causal in nature. ”

---

## [Decision Letter · Decision Letter 1]

19 Jun 2023

“Help! I need some music!”: Analysing music discourse & depression on Reddit

PONE-D-23-04461R1

Dear Dr. Singh,

We’re pleased to inform you that your manuscript has been judged scientifically suitable for publication and will be formally accepted for publication once it meets all outstanding technical requirements.

Kind regards,

Michal Ptaszynski, PhD

Academic Editor

PLOS ONE

Additional Editor Comments (optional):

Reviewers' comments:

Reviewer's Responses to Questions

**Comments to the Author**

1. If the authors have adequately addressed your comments raised in a previous round of review and you feel that this manuscript is now acceptable for publication, you may indicate that here to bypass the “Comments to the Author” section, enter your conflict of interest statement in the “Confidential to Editor” section, and submit your "Accept" recommendation.

Reviewer #1: All comments have been addressed

2. Is the manuscript technically sound, and do the data support the conclusions?

Reviewer #1: Yes

3. Has the statistical analysis been performed appropriately and rigorously? 

Reviewer #1: Yes

4. Have the authors made all data underlying the findings in their manuscript fully available?

Reviewer #1: Yes

5. Is the manuscript presented in an intelligible fashion and written in standard English?

Reviewer #1: Yes

6. Review Comments to the Author

Reviewer #1: Dear Authors,

Thank you for taking the time to edit the paper and good luck with the final steps of the preparation of the manuscript.

I'll leave it to the editor to advise on the journal's policy regarding how the quote in the title should be presented, given it's a nod to the famous song rather than a specific quote.

7. PLOS authors have the option to publish the peer review history of their article (what does this mean?). If published, this will include your full peer review and any attached files.

Reviewer #1: No

---

## [Editor Report · Acceptance letter]

12 Jul 2023

PONE-D-23-04461R1 

“Help! I need some music!”: Analysing music discourse & depression on Reddit 

Dear Dr. Singh:

I'm pleased to inform you that your manuscript has been deemed suitable for publication in PLOS ONE. Congratulations! Your manuscript is now with our production department. 

Kind regards, 

on behalf of

Dr. Michal Ptaszynski 

Academic Editor

PLOS ONE